# The First Wave of the COVID-19 Pandemic Strengthened the “Strong” and Weakened the “Weak” Ones

**DOI:** 10.3390/ijerph192114523

**Published:** 2022-11-05

**Authors:** Albertas Skurvydas, Ausra Lisinskiene, Daiva Majauskiene, Dovile Valanciene, Ruta Dadeliene, Natalja Istomina, Ieva Egle Jamontaite, Asta Sarkauskiene

**Affiliations:** 1Department of Rehabilitation, Physical and Sports Medicine, Faculty of Medicine, Institute of Health Scences, Vilnius University, 21/27 M.K. Čiurlionio St., 03101 Vilnius, Lithuania; 2Education Academy, Vytautas Magnus University, K. Donelaičio Str. 58, 44248 Kaunas, Lithuania; 3Institute of Education Studies, Education Academy, Vytautas Magnus University, K. Donelaičio Str. 58, 44248 Kaunas, Lithuania; 4Institute of Health Sciences, Faculty of Medicine, Vilnius University, 21/27 M.K. Čiurlionio St., 03101 Vilnius, Lithuania; 5Department of Sports Recreation and Tourism, Klaipėda University, HerkausMantost. 84, 92294 Klaipėda, Lithuania

**Keywords:** physical activity, MVPA, BMI, health, depression, stress, impulsivity, adults

## Abstract

The aim of this study was to explore how the first wave of the COVID-19 pandemic, during which contact communication was severely restricted, changed psychological health indicators, such as subjective assessment of health and depression, impulsivity, stress and emotional intelligence (EI) and how that depended on age, gender, physical activity (PA), sports specificity and body mass index (BMI).We surveyed 6369 before and 2392 people during the first wave of the COVID-19 pandemic. The participants were aged 18–74 years. Participants completed the Danish Physical Activity Questionnaire (DPAQ), the 10-item Perceived Stress Scale (PSS-10), the Schutte Self-Report Emotional Intelligence Test (SSREIT), Barratt Impulsiveness Scale Version 11 (BIS-11), subjective depression and health self-assessments. One-way and two-way analyses of variance (ANOVA) were performed to assess the effect of independent variables on the dependent variables of MVPA (METs). Statistical analysis showed that restrictions during the first wave of the COVID-19 pandemic did not alter moderate-to-vigorous physical activity (MVPA), except for a significant decrease in MVPA in women aged 18–25 years, or body mass index in women and men of different ages. An increase in depression and impulsivity was observed, especially an increase in unplanned or spontaneous activity. The restrictions during the first wave increased stress in women of all ages and, rather unexpectedly, improved health self-assessment in men.The study showed that the first wave of the COVID-19 pandemic affected people’s subjective assessment of health, depression, stress and impulsivity in two ways: it “weakened the weak ones” and “strengthened the strong ones”.

## 1. Introduction

Evidence-based research increasingly suggests that various forms and doses of physical activity (PA) are effective in combating many chronic diseases [1,2], improving well-being and mental health [3,4,5,6] and reducing all-cause mortality [7]. The effect of PA on various body functions is very specific and depends in a non-linear manner on the intensity, duration and load in muscle work [2,8]. In addition, the health benefits of PA also depend on the age, gender, health status and body mass index (BMI) of people [2,3,7,9]. Insufficient PA has been shown to increase obesity and subsequently lead to systemic inflammation, which causes many chronic diseases [1,10]. Guthold, Stevens, Riley and Bull summarised the dynamics of physical inactivity in 1.9 million people between 2001 and 2016 in developed European countries and found that it increased significantly in both women and men [11]. Obesity and low PA can be intertwined, i.e., low PA leads to obesity, and obesity reduces the motivation to be physically active [12]. Evidence from epidemiological studies suggests that there is a strong bidirectional relationship between depression and obesity, i.e., BMI increases the risk of developing depression, and conversely, individuals with depression are at increased risk for high BMI [13,14,15].

During the first wave of the coronavirus disease 19 (COVID-19) pandemic, the closure of gyms and indoor athletic and leisure centres, the cancellation of recreational sports and restrictions on all but essential travel likely caused a decline in the amount of PA performed by people [16,17]. A study showed an increase in body weight in about half of the participants during the first COVID-19 pandemic period in Poland, which was associated with a decrease in PA and an increase in consumption of total food and high-energy-density products [18]. A similar conclusion was reached by other researchers who found that the COVID-19 pandemic increased snack and alcohol consumption [19].These studies showed that the COVID-19 pandemic has affected healthy lifestyles, the main components of which are often considered healthy eating, PA, adequate sleep, limited tobacco and alcohol consumption, stress management and maintenance of relationships [20]. However, other studies found that some participants reported an increase in healthy lifestyle habits since the start of the pandemic in the United States (i.e., 36% reported an increase in healthy eating habits and 33% reported an increase in PA) [21]. However, they also reported increases in addictive lifestyle habits, including alcohol use (40%), tobacco use (41%) and vaping (46%).The results of our recent study show that during COVID-19, ‘good stress’ was represented in Lithuania, which mobilised people to exercise more individually in sports clubs, overeat less and not start consuming more alcohol [15].

Changes in diet, sleep quality and PA were found to be related to differences in negative mood during the COVID-19 lockdown [22]. Preliminary evidence suggests that symptoms of anxiety and depression (16–28%) and self-reported stress (8%) are common psychological reactions to the COVID-19 pandemic [23]. PA increased positive mood in college students regardless of stressful life events during the pandemic [24]. Moreover, social isolation during the pandemic had a negative effect on people’s psychological well-being [25]. However, PA reduced the symptoms of depression and anxiety during the pandemic [26].

Despite the studies mentioned above, it is not yet clear how the first wave of the COVID-19 pandemic, during which contact communication was severely restricted, changed psychological health indicators, such as subjective assessment of health and depression, impulsivity, stress and emotional intelligence (EI), and how that depended on age, gender, PA, sports specificity and BMI. We proposed the following basic hypothesis: Restrictions during the first wave of the COVID-19 pandemic should worsen psychological health indicators more than BMI and PA. In addition, deterioration in health, as well as an increase in depressive symptoms, should be more pronounced in the elderly and overweight people, who have the least PA and do not exercise on their own, are the moststressed and impulsive in decisionmaking, and have the least EI. The aim of this study was to explore how the first wave of the COVID-19 pandemic, during which contact communication was severely restricted, changed psychological health indicators, such as subjective assessment of health and depression, impulsivity, stress and EI, and how that depended on age, gender, PA, sports specificity and BMI.

## 2. Methods

### 2.1. Participants

We surveyed 6369 people (4545 women and 1824 men) before and 2392 people (1856 women and 536 men) during the COVID-19 pandemic. The participants were aged 18–74 years. The proportion of people with higher and university education in the samples before and during the pandemic was 78.3% and 79.5%, respectively. The urban population constituted 83.4% and 83.1% of the samples before and during the pandemic, respectively. The ages of the participants in these two groups were 37.9 ± 11.8 years and 38.4 ± 12.6 years, respectively.

Using the survey, we determined the BMI and specificity of PA of the participants, i.e., what percentage of participants did not exercise, were professional athletes, exercised independently and exercised at sports and health centres. Moreover, we determined psychological health indicators, emotional intelligence and impulsivity.

The study was conducted from October 2019 to June 2020 and from November 2020 to March 2021. Participants were selected from Lithuania to represent a Lithuanian sample. Participation was anonymous, and data collection and processing were confidential. We used an online survey to collect information using Google Forms (https://docs.google.com/forms/ (accessed on 6 March 2021)). All participants completed the online questionnaires. The online survey was distributed by the researchers via social media (Facebook) and personal messages (WhatsApp). Using the survey, we determined the BMI and specificity of PA of the participants, i.e., what percentage of participants did not exercise, were professional athletes, exercised independently and exercised at sports and health centres.

Ethics committee approval to conduct this study was obtained from Klaipeda University. We ensured that the study was conducted in accordance with other ethical documents [27,28].

### 2.2. Measures

Danish Physical Activity Questionnaire (DPAQ). The DPAQ was adapted from and differs from the International Physical Activity Questionnaire in that it refers to PA in the past 24 h (for 7 consecutive days) rather than the past 7 days. The selected activities were listed on the PA scale in nine levels of physical exertion in metabolic equivalents (METs), ranging from sleep or inactivity (0.9 MET) to very strenuous activity (> 6 METs). Each level (A = 0.9 MET, B = 1.0 MET, C = 1.5 METs, D = 2.0 METs, E = 3.0 METs, F = 4.0 METs, G = 5.0 METs, H = 6.0 METs, and I > 6 METs) was described with examples of specific activities at each MET level and a small drawing. The PA scale was constructed to fill in the number of minutes (15, 30, or 45) and hours (1–10) spent at each MET activity level in an average 24 h weekday. This allowed the calculation of total MET time, which includes 24 h of sleep, work and leisure time on an average weekday [29,30].

We calculated how much energy (in METs) was expended per day during sleep, sedentary (0.9–1.5 METs), light-intensity PA (1.5–3 METs), moderate-intensity PA (MPA; 3–6 METs) and vigorous-intensity PA (VPA; >6 METs). We also combined MPA with VPA as moderate-to-vigorous PA (MVPA). Moreover, in this study, we only used the MVPA indicator.

Subjective health self-assessment. For this purpose, the following 4-point scale was used: poor health (1 point), satisfactory health (2 points), good health (3 points) and excellent health (4 points).

Perceived stress and depression. The 10-item Perceived Stress Scale (PSS-10) was used to measure participants’ stress levels [31]. In the PSS-10, participants are asked to answer 10 questions about their feelings and thoughts during the past month on a 5-point scale ranging from 0 to 4. Higher scores indicate higher levels of perceived stress.

Subjective depression self-assessment.Each item was assessed on a four-point (0–3) response category, namely: was not overwhelmed by depression (0 points), depression was more prevalent than before (1 point), depression was prevalent slightly more frequently than before (2 points) anddepression experienced much more often than before (3 points).

Assessment of EI. This was performed using the Schutte Self-Report Emotional Intelligence Test (SSREIT) [32]. This test is a 33-item questionnaire divided into four subscales, which are: perception of emotions (10 items), ability to deal with one’s own emotions (9 items), ability to deal with the emotions of others (8 items) and use of emotions (5 items). The items are designed to be answered on a 5-point scale ranging from 1 (strongly disagree) to 5 (strongly agree). Total scores range from 33 to 165, with higher scores indicating greater ability in the area of EI.

Assessment of impulsivity. Impulsiveness was assessed using the Barratt Impulsiveness Scale version 11 (BIS-11) [33]. This scale is a 30-item questionnaire divided into three subscales: attentional impulsivity, scored with eight items; motor impulsivity, scored with 11 items; and non-planning impulsivity, scored with 11 items. Items are answered on a 4-point scale ranging from 1 (rarely/never) to 4 (almost always/always). Total scores range from 30 to 120, with higher scores representing higher impulsivity.

### 2.3. Statistical Analyses

Interval data are given as mean± standard error. All interval data were confirmed to be normally distributed using the Kolmogorov–«Smirnov test. To determine if there was an interaction between the independent variables and the dependent variable, two- and three-way ANOVA were performed. When significant effects were found, Tukey’s post hoc adjustment was used for multiple comparisons within each repeated measure ANOVA. The partial eta squared value (${ŋ}_{P}^{2}$) was estimated as a measure of effect size, and chi-square (χ^2^) tests were performed. For all tests, statistical significance was defined as *p <* 0.05. Statistical analyses were performed using IBM SPSS Statistics software (version 22; IBM SPSS, Armonk, NY, USA).

## 3. Results

### 3.1. Effect of the COVID-19 Pandemic on Psychological Health Indicators

Interestingly, men aged 18–25 years and 26–44 years performed more frequent health self-assessment during the pandemic than before it (interaction effect of COVID-19 pandemic × age × gender: *p* = 0.001, ${ŋ}_{P}^{2}$ = 0.001, Figure 1).

The pandemic had no significant effect on women’s health self-assessment (χ^2^ = 5.2; *p* = 0.157; Table 1).

The COVID-19 pandemic significantly increased depressive symptoms in both women and men (COVID-19 pandemic effect: *p <* 0.0001, ${ŋ}_{P}^{2}$ = 0.003; age effect: *p <* 0.0001, ${ŋ}_{P}^{2}$ = 0.013; gender effect: *p <* 0.0001, ${ŋ}_{P}^{2}$ = 0.005; interaction effect: nonsignificant). However, it increased impulsivity (COVID-19 pandemic effect: *p <* 0.0001, ${ŋ}_{P}^{2}$= 0.005; age effect: *p* = 0.014, ${ŋ}_{P}^{2}$ = 0.003; gender effect: *p* = 0.19, ${ŋ}_{P}^{2}$ = 0.001; interaction effect: nonsignificant) and stress only in women (interaction effect of COVID-19 pandemic × gender: *p* = 0.024, ${ŋ}_{P}^{2}$ = 0.001) (Figure 2). In particular, the pandemic increased non-planning impulsivity for both women and men (COVID-19 pandemic effect: *p <* 0.0001, ${ŋ}_{P}^{2}$ = 0.018; age effect: *p* = 0.324, ${ŋ}_{P}^{2}$= 0.003; gender effect: *p* = 0.795, ${ŋ}_{P}^{2}$ < 0.0001; interaction effect: nonsignificant). EI did not change significantly due to the pandemic in either women or men (COVID-19 pandemic effect: *p <* 0.53, ${ŋ}_{P}^{2}$ < 0.0001; age effect: *p <* 0.0001, ${ŋ}_{P}^{2}$ = 0.004; gender effect: *p <* 0.0001, ${ŋ}_{P}^{2}$ = 0.006; interaction effect: nonsignificant). Thus, the COVID-19 pandemic improved health self-assessment in men, increased stress in women and increased depression and impulsivity in women and men of all ages.

### 3.2. Effect of EI, Stress, MVPA, BMI, Sports Specificity and Impulsivity on Changes in Health Due to the COVID-19 Pandemic

The pandemic led to an improvement in the health of both women and men with higher EI. However, the health scores worsened in those with lower EI (interaction effect of COVID-19 pandemic × EI: *p* < 0.0001, ${ŋ}_{P}^{2}$ = 0.004) (Figure 3). The deterioration in health due to the pandemic increased as the stress increased in both women and men (*p* < 0.0001, ${ŋ}_{P}^{2}$ = 0.009). However, in the case of low stress, the pandemic even led to an improvement in the health self-assessment of both women and men (*p* < 0.001). We found that the lower the incidence of MVPA (for both women and men), the greater the deterioration in health due to the pandemic. Moreover, the pandemic even led to an improvement in the health of the most physically active people (interaction effect of COVID-19 pandemic × MVPA: *p <* 0.0001, ${ŋ}_{P}^{2}$ = 0.002). The COVID-19 pandemic’s effects on health also depended on BMI (interaction effect of COVID-19 pandemic × BMI for health self-assessment: *p* = 0.002, ${ŋ}_{P}^{2}$ = 0.002). We found that the higher the level of obesity (for both women and men), the greater the deterioration in health due to the pandemic, except for men with low BMI, for whom health deterioration due to the pandemic was also observed. Our findings showed that the health of people who did not exercise (both women and men) deteriorated due to the COVID-19 pandemic compared with those who exercised (interaction effect of COVID-19 pandemic × exercise for health: *p <* 0.0001, ${ŋ}_{P}^{2}$ = 0.018). We found that the greater the depression in both women and men, the greater the deterioration in health due to the pandemic (interaction effect of COVID-19 pandemic × depression for health: *p <* 0.0001, ${ŋ}_{P}^{2}$ = 0.013). However, the change in health self-assessment during the pandemic did not depend on the subjects’ impulsivity (interaction effect of COVID-19 pandemic × impulsivity for health: *p* = 0.66, ${ŋ}_{P}^{2}$ < 0.0001).

### 3.3. Effect of Health on Changes in Depression, Stress and Impulsivity Due to the COVID-19 Pandemic

We found that the worse the health of women and men, the greater the increase in depression (interaction effect of COVID-19 pandemic × health for depression: *p <* 0.0001, ${ŋ}_{P}^{2}$ = 0.012) and stress (interaction effect of COVID-19 pandemic × health for stress: *p <* 0.0001, ${ŋ}_{P}^{2}$ = 0.018) due to the pandemic (Figure 3). However, health self-assessments did not have a significant effect on the increase in impulsivity in both women and men during the pandemic (interaction effect of COVID-19 pandemic ×health for impulsivity: *p* = 0.23, ${ŋ}_{P}^{2}$ = 0.002), although impulsivity increased due to the pandemic in women and men with the worst health.

### 3.4. Effect of the COVID-19 Pandemic on MVPA and BMI

The number of people who did not exercise (both women and men) did not change because of the pandemic, but a redistribution occurred: the number of people who exercised in sports centres decreased significantly (*p <* 0.001), and the number of people who exercised on their own increased (women: *p <* 0.001; χ^2^ = 400.1; men: *p <* 0.001; χ^2^ = 67.5). The proportion of women and men who did not exercise was 21.6% and 38% before the COVID-19 pandemic and 25.7% and 39.2% during the pandemic, respectively. The proportion of women and men who exercised independently was 48% and 29.6% before the pandemic and 58% and 49.5% during the pandemic, respectively. The proportion of women and men in a professional sport was 7.6% and 3.4% before the pandemic and 9.3% and 2.6% during the pandemic, respectively. A total of 21.6% and 38% of women and men exercised in sports clubs before the pandemic and 6.9% and 8.7% during the pandemic, respectively.

The COVID-19 pandemic did not significantly change MVPA (COVID-19 pandemic effect: *p* = 0.076, ${ŋ}_{P}^{2}$= 0.001; age effect: *p <* 0.0001, ${ŋ}_{P}^{2}$ = 0.005; gender effect: *p <* 0.0001, ${ŋ}_{P}^{2}$ = 0.02; interaction effect: nonsignificant), although the pandemic significantly reduced MVPA in young women (*p <* 0.001; Figure 4). Rather unexpectedly, the pandemic did not significantly change BMI in either women or men (COVID-19 pandemic effect: *p* = 0.8, ${ŋ}_{P}^{2}$ < 0.0001; age effect: *p <* 0.0001, ${ŋ}_{P}^{2}$ = 0.036; gender effect: *p <* 0.0001, ${ŋ}_{P}^{2}$ = 0.023; interaction effect: nonsignificant; Figure 5).

## 4. Discussion

Our findings clearly showed that although the restrictions during the first wave of the COVID-19 pandemic did not alter either MVPA (except for a significant decrease in the MVPA of women aged 18–25 years) or BMI in women and men of different ages, there was an increase in depression and impulsivity, especially an increase in unplannedor spontaneous activity. Moreover, the restrictions during the first wave increased stress in women of all ages and, unexpectedly, improved health self-assessment in men. Thus, the first wave of the COVID-19 pandemic worsened health in those whose EI was <125, VMPA < 10 METs, BMI > 30 kg/m^2^ and PSS > 26. In addition, poor health contributed to the greatest increase in depression and impulsivity during the pandemic. To our knowledge, this is the first study to show that the first wave of the COVID-19 pandemic affected people’s subjective assessment of health, depression, stress and impulsivity in the following two ways: people who did not have a sufficiently healthy lifestyle and felt that they had “poor health” became even weaker, and those who lived an active and healthy life and felt healthy before the pandemic became stronger. In addition, our findings showed that psychological changes in health indicators during the COVID-19 pandemic were independent of age.

This study has clearly shown that the first wave of the COVID-19 pandemic was “good stress” that prompted both women and men to exercise more independently and not reduce MVPA (except for young women), which is the most important guarantor of enhancing one’s health [2,34]. Zhang et al. reported similar findings suggesting that the COVID-19 pandemic increased healthy eating behaviours and PA in the United States [21]. The finding that MVPA did not change during the pandemic in our study is a good “sign” of well-being. Other studies showed that social isolation during the pandemic had a negative effect on people’s psychological well-being [25,35]. However, PA during the pandemic reduced the symptoms of depression and anxiety [26]. This finding is consistent with the data from our study that the psychological health improved more among people who did not exercise at all to the intensity of MVPA and those who did not exercise independently. Another “good sign” in our study is that an increase in BMI, which increases the risk for severe COVID-19 [36], was not observed during the pandemic.

Our findings are inconsistent with the data from other studies indicating that BMI increased in people due to social isolation during the pandemic [18]. Evidence from epidemiological studiesindicated that depression and obesity have a strong bidirectional relationship, i.e., BMI increases the risk of developing depression, and, vice versa, individuals with depression have an increased risk of high BMI [13]. This finding is consistent with our study suggesting that deterioration in health outcomes was the worst among participants (both women and men) who were obese. We believe that the increase in EI with age may have “compensated” for the negative effects of an increase in BMI on the subjective assessment of health, depression and stress. It appears that some people experienced a sudden increase in being overweight and obese, but because of their EI, as well as other reasons, they did not feel that it was “bad for their health.” This finding is consistent with the data from other studies indicating that various social and psychological factors have a positive effect on people’s well-being [37]. In addition, we found that the COVID-19 pandemic improved health among people with the highest EI.

The main limitation of our studies is that the same research subjects were not researched before and during the COVID-19 pandemic (some of the research subjects may have been the same ones, but we were unable to trace them). Thus, we were unable to determine any mechanistic causes between the dependent and independent variables. For this reason, it is quite difficult to determine what the cause is and what the consequenceswere. We did not have the possibility to ascertain the experience of thoseresearched (their families) in coping with stress that was undoubtedly caused by theCOVID-19 pandemic. Moreover, we do not know if they applied any techniques for coping with stress (for example, mindfulness) and, if so, which ones.

## 5. Conclusions

We found that the first wave of the COVID-19 pandemic affected more changes in depression and impulsivity (especially in the increase in unplanned or spontaneous activity) than in the MVPA and BMI of women and men of different ages, except for a significant decrease in the MVPA of women aged 18–25 years. The restrictions during the first wave of the COVID-19 pandemic increased stress in women of all ages and, unexpectedly, improved health self-assessments in men. Most importantly, the first wave of the pandemic worsened the health of people with the lowest EI and MVPA, the highest BMI and stress levels and those who did not exercise. In addition, poor health contributed to the greatest increase in depression and impulsivity during the pandemic. To our knowledge, this is the first study showing that the first wave of the COVID-19 pandemic affected people’s subjective assessment of health, depression, stress and impulsivity in two ways, it “weakened the weak ones” and “strengthened the strong ones”.

## Figures and Tables

**Figure 1 ijerph-19-14523-f001:**
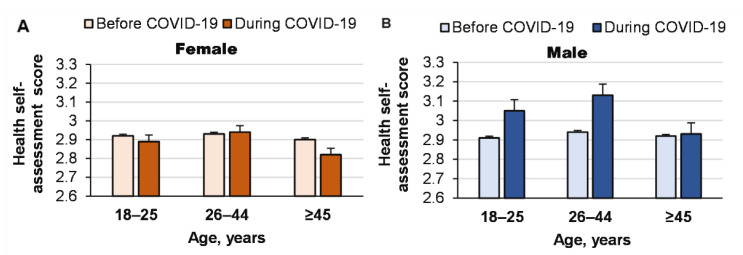
Influence of COVID-19 on health self-assessments of men and women of different ages.

**Figure 2 ijerph-19-14523-f002:**
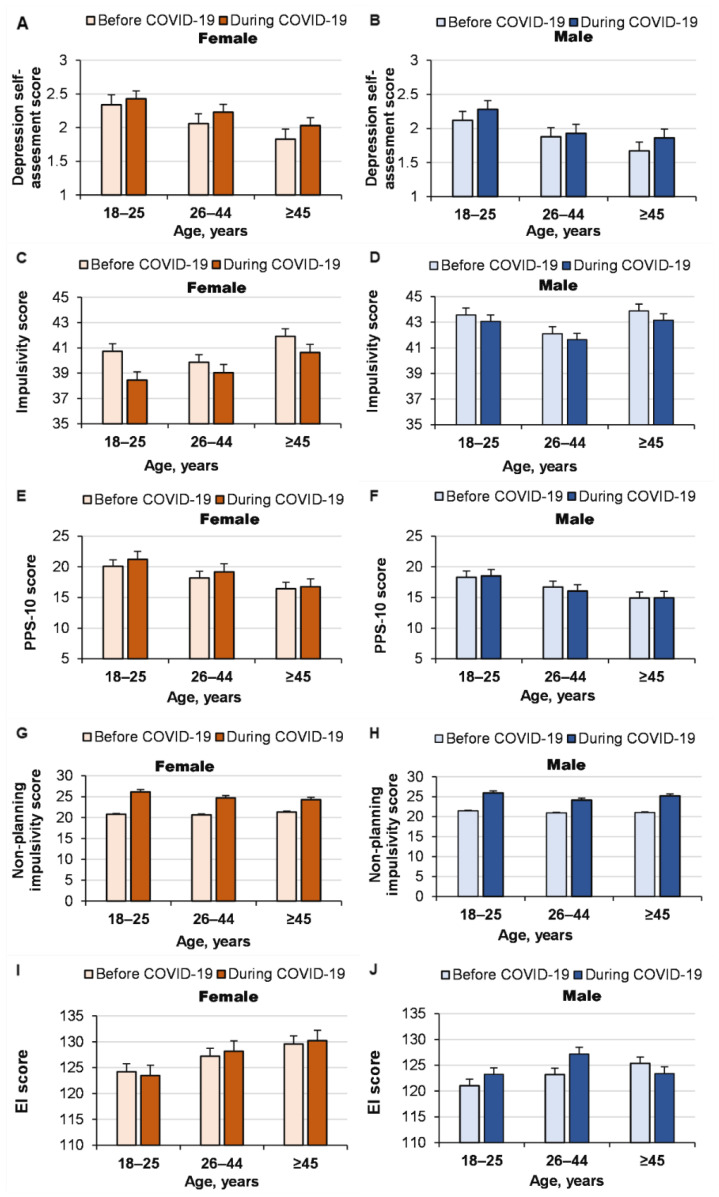
Effect of the COVID-19 pandemic on depression (**A**,**B**), impulsivity (**C**,**D**), stress (evaluated using the PSS-10) (**E**,**F**), non-planning impulsivity (**G**,**H**) and EI (**I**,**J**) in women and men of different ages.

**Figure 3 ijerph-19-14523-f003:**
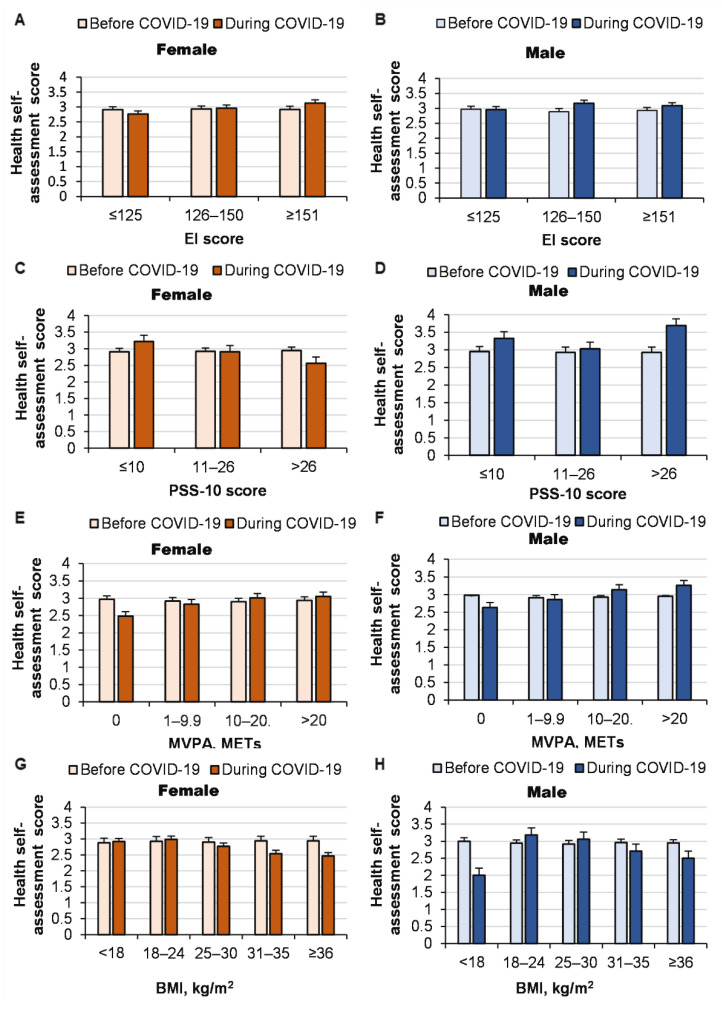
Health self-assessments before and during the COVID-19 pandemic according to EI (**A**,**B**), stress (evaluated using the PSS-10) (**C**,**D**), MVPA (**E**,**F**), BMI (**G**,**H**), sports specificity (**I**,**J**), impulsivity (**M**,**N**) and depression self-assessments (**K**,**L**).

**Figure 4 ijerph-19-14523-f004:**
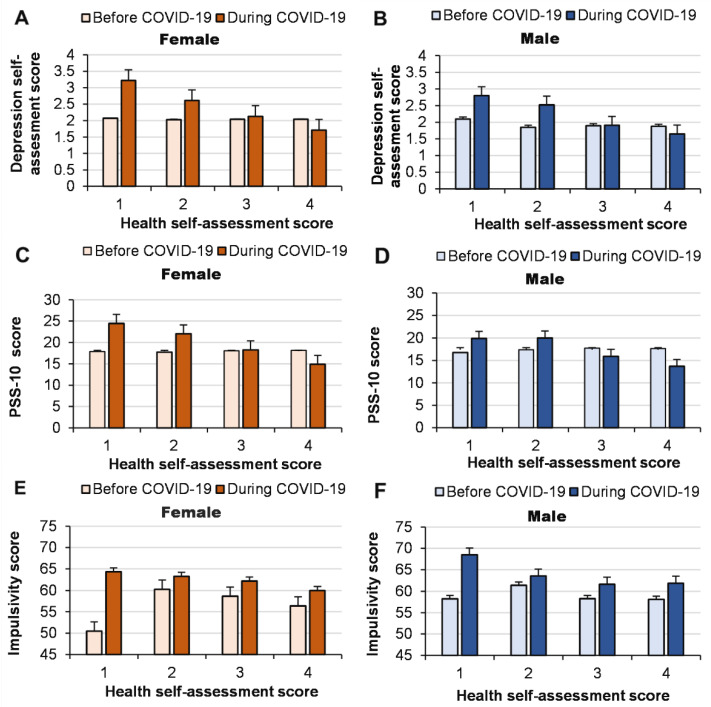
Depression self-assessments (**A**,**B**), stress (evaluated the PSS-10) (**C**,**D**) and impulsivity (**E**,**F**) before and during the COVID-19 pandemic according to health.

**Figure 5 ijerph-19-14523-f005:**
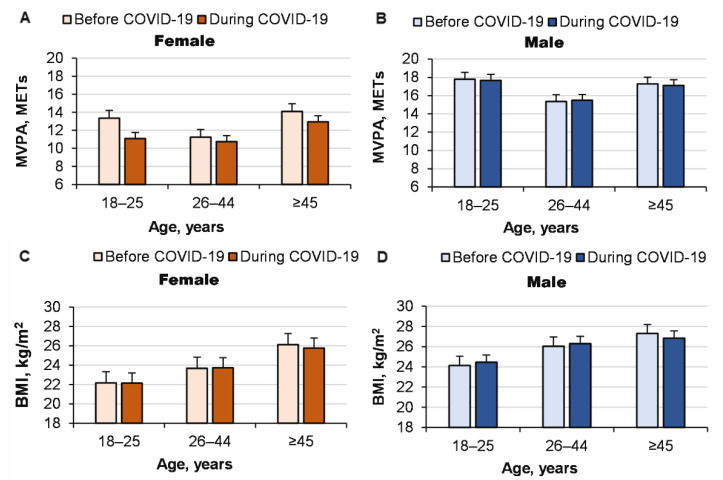
MVPA (**A**,**B**) and BMI (**C**,**D**) before and during the COVID-19 pandemic.

**Table 1 ijerph-19-14523-t001:** Influence of the pandemic on the health self-assessments of women and men.

		Female				Male			
Variables of Health		Before COVID-19	During COVID-19	Chi-Square	*p*-Value	Before COVID-19	During COVID-19	Chi-Square	*p*-Value
Poor	Count	104_a_	36_a_			48_a_	10_a_		
	%	2.3%	2.0%			2.6%	1.9%		
Satisfactory	Count	1020_a_	446_a_			392_a_	89_b_		
	%	22.4%	24.3%			21.5%	16.7%		
Good	Count	2551_a_	1040_a_	5.2	0.157	1020_a_	293_a_	13.3	0.004
	%	56.2%	56.5%			55.9%	55.1%		
Excellent	Count	869_a_	316_a_			364_a_	140_b_		
	%	19.1%	17.2%			20.0%	26.3%		

Each subscript letter denotes a subset of before and during COVID-19 categories whose column proportions do not differ significantly from each other at the *p* < 0.05 level. Each subscript letter denotes a subset of COVID-19 categories whose column proportions are not significantly different at the *p* = 0.05 level.

## Data Availability

The data presented in this study are available on request from the corresponding author.

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
