# Peer review of "The First Wave of the COVID-19 Pandemic Strengthened the “Strong” and Weakened the “Weak” Ones"

_ijerph, 2022, doi:10.3390/ijerph192114523_

Round 1
Reviewer 1 Report
The presented manuscript analyses an interesting topic of high relevance in years of COVID and post-COVID. It incorporates different aspects related to physical activity levels, emotional and mental health issues, and changes due to the COVID-19 pandemic. The selected topic and obtained results are very important in years dealing with pandemics in terms to identify changes that occur and defining strategies on how to cope with it and help people to maintain their health and well-being.
The manuscript is written in a clear and easy-to-understand manner, having a well-defined structure. The introduction is in line with the research problem and has a good number of relevant references. The method of work is properly selected. Obtained results are presented extensively, that in certain point is difficult to follow, yet acceptable, due to the many variables included in the study. The discussion section can be improved by adding possible reasons and explanations for obtained results and not just cooperation with similar results from other studies.
In terms of improving the quality of the presented manuscript, I suggest the following changes to be done:
- introduction section represents study results from different countries such as Poland, the USA etc. Considering that the study sample is adults from Litvania, it will be good to share results from the Litvanian context during a pandemic. Some studies are presented in the reference list and their results can be emphasized here,
- The aim of the study is presented in the abstract but not in the main text. There only two hypotheses are presented. I suggest, aim to be added in the main text as well (Line 78 - 88).
- Line 91 -to define clearly what period before COVID and what period during COVID the study took place
-Line 92. Will be better to provide more information about the structure of the study sample (distribution per age, gender etc in the sample before and after COVID)
102 -105. To be adjusted based on all variables included in the survey. Only variables for PA level are described
Line 106 - 107. Please add the name of the university that provided the Ethical approval
Line 136. Please explain the abbreviation EI (in case if I missed that is explained somewhere else previously)
Reliability and validity for the Litvanian sample of applied instruments are not presented. Please provide this information.
Line 276 - 280 Please re-write and explain better the text in the suggested lines.
The discussion section can be supplemented with a deeper explanation of possible reasons for changes identified with the study, not just with simple cooperation with results from previous studies.
Wish you success!
Author Response
Dear Reviewer,
Thank you very much for the your’s comments concerning our manuscript titled: The first wave of the COVID-19 pandemic strengthened the “strong” and weakened the “weak” ones. Those comments are all valuable and very helpful for revising and improving our paper. We have studied comments carefully and have made correction which we hope will meet with approval. Revised portion are marked in the paper.

Reviewer 2 Report
The issue raised is of interest to assess the coping of the pandemic experienced and its impact on the psychological and physical well-being of the population. Studies that report on these facts and consequences are useful to better prepare the population and increase their resilience.
However, in this study, the associations established between the different variables related to mental health and psychological well-being did not consider the effect of the occurrence of COVID-19 deaths on family members and close relatives.
The occurrence of losses in the family circle is undoubtedly a factor that explains alterations in the measurements of the psychological variables evaluated and which unfortunately has been a frequent reality during the pandemic. I think you should bear these observations in mind when drafting your conclusions.
Given the type of study and design used, it would be advisable to avoid terms such as "testing hypotheses".
It is more appropriate to refer to terms such as: objectives being pursued and implications arising from the objectives of the study.
It is not clear to me how the BMI was calculated for each individual:
Was it a specific question included in a questionnaire?
Or was it calculated based on the weight and height reported by each respondent?
An important aspect: They should provide data on the available reliability and validity of the assessment instruments used. They should also indicate the age range and type of population for which they were originally validated.
The sample is of considerable size and the use of the proposed analyses is correct.
Author Response

(The authors gave the same response as above.)
